# Performance of Concrete Mixes Containing TBM Muck as Partial Coarse Aggregate Replacements

**DOI:** 10.3390/ma14216263

**Published:** 2021-10-21

**Authors:** Ala Abu Taqa, Mohamed Al-Ansari, Ramzi Taha, Ahmed Senouci, Ghaleb M. Al-Zubi, Mohamed O. Mohsen

**Affiliations:** 1Department of Civil and Architectural Engineering, Qatar University, Doha P.O. Box 2713, Qatar; m.alansari@qu.edu.qa (M.A.-A.); 200202128@student.qu.edu.qa (M.O.M.); 2Engineering Program, Schreiner University, Kerrville, TX 78028, USA; ramziabdtaha@gmail.com; 3Department of Construction Management, University of Houston, Houston, TX 77204-4020, USA; asenouci@central.uh.edu; 4Arab Center for Engineering Studies, Doha P.O. Box 19579, Qatar; gzubi@aces-int.com

**Keywords:** TBM muck, coarse aggregates, recycling, sustainability, compressive strength, flexural strength, EDX analysis, SEM images

## Abstract

This study investigated the potential utilization of the TBM muck obtained from the Gold Line of the Doha Metro Project as a partial replacement of coarse aggregates in concrete mixes. First, the TBM muck particles were screened to coarse aggregate standard sizes. Then, concrete mixes were prepared using 0%, 25%, 50%, and 75% TBM muck replacement of coarse aggregates. The compressive and flexural strengths were determined for all mixes at 28 and 56 days. Moreover, the results obtained were validated using EDX analysis and SEM images. A *t*-statistical analysis did not show a significant impact of TBM muck usage on the compressive strength results of the concrete mixes. However, another *t*-statistical analysis showed that TBM muck replacement of coarse aggregates had adversely affected the flexural strength results. The EDX analysis indicated the presence of Na^+^ ions, which can replace the Ca^2+^ ions in the C-S-H gel, cause discontinuities of it, and hence reduce the strength at later ages. Finally, the SEM images showed that the ettringite and carbon hydroxide (C-H) contents in the mixes with TBM muck were higher than that of the control mix, while the C-S-H gel was less in such mixes.

## 1. Introduction

### 1.1. General

With the population and economy boom witnessed globally, the demand for new buildings, structures, and effective infrastructural systems has considerably increased. Constructing underground highway networks and railway tunnels above all is the most suitable method to adapt to such development. Instead of conventional blast and drill excavation methods, Tunnel Boring Machines (TBMs) have been recently used in many countries to excavate tunnels in hard rocks. It has been also demonstrated that the TBM technology is a feasible method for the excavation of underground research laboratories (URLs), which play important roles in the safe disposal of high-level radioactive waste (HLW) [1]. The TBM machines produce millions of tons of muck every year, which are mostly stockpiled as waste disposal.

The consumption of non-renewable natural raw materials has recently become a challenge. TBM muck use presents a viable utilization for sustainable economic and environmental developments. However, TBM muck must achieve certain physical and chemical properties to be used as a building material. Several studies investigated the potential use of TBM muck in construction applications. Few of these studies concluded that TBM could be used in many specialized building applications as an aggregate substitute with less processing durations [2,3]. Few others indicated that rock formation muck, which possesses low-to-medium brittleness and medium-to-high hardness, had the highest potential to be used in road and building construction [4,5,6,7,8,9]. Grünner et al. [6] evaluated the muck extracted from a Mismove tunnel in northern Slovakia using a set of laboratory tests including adsorption and short frost resistance tests. They concluded that muck’s use must be evaluated based on its features, extraction area geology, and driving methodology. Thus, a final decision was not possible to be made on the use of this muck in different construction applications. Bellopede and Marini [7] investigated the influence of treatment on the muck that was extracted using TBM as well as conventional drilling and blast excavation methods. They observed that raw TBM muck particles are flaky in shape and therefore not suitable to replace concrete aggregate in construction applications. They observed that mobile and fixed plants-treated TBM muck by a hammer crusher and a secondary jaw crusher can reduce the muck flakiness index to conform to concrete aggregate specifications. Berdal et al. [9] investigated the use of TBM fines, with sizes ranging between 0 and 0.125 microns from two Norwegian projects, in concrete production. They concluded that TBM fines had a better effect on concrete rheology as well as better flow properties compared to natural coarse aggregates. Thalmann-Suter [4] conducted several laboratory tests such as point load, breakability, and petrographic tests to investigate the properties of the TBM muck that was extracted from the AlpTransit tunnelling project in Switzerland. They reported that the TBM muck is suitable for use as concrete aggregates after adequate preparation and special processing. Berdal [8] observed that the properties of TBM muck are largely affected by the area geology and extraction technology used. Therefore, extracted muck from each project must be studied separately to determine its appropriate processing requirements. Gertsch et al. [5] analyzed a laboratory boring sample from the Yucca Mountain project site in the USA, which was obtained by fitting five boulders into a laboratory tunnel boring machine (LTBM) and crushing them. The sample was composed of well-graded hard rocks with flaky chips and little fines. The authors concluded that TBM muck could be successfully used as a fill or sub-grade material in special construction projects, with little processing of the TBM muck. Moreover, they reported on a few tunneling projects, where their extracted TBM muck was used in construction applications, such as for fill material, road base and/or sub-base, and landscaping. However, the muck that was extracted from the Jostedal hydropower project in Norway was used in Portland cement concrete along with 30% natural gravel. The produced concrete mixes exhibited large variations in the compressive strength and concrete quality because the TBM muck lacked quality control as it was taken blindly from stockpiles without consideration of its properties. Moreover, the authors also reported that differences in the untreated and uncharacterized muck particle-grading, shape, and mechanical strength contributed to the large performance variations. Hence, the muck should be carefully characterized before using it in concrete mixes [5].

Recently, in the 14th Baltic Sea Region Geotechnical Conference, Alnuaim [10] presented several potential areas to utilize the TBM material excavated from the Riyadh Metro project. After investigating the grain size distribution, Atterberg limits, compaction test, and direct shear and permeability tests, the author concluded that utilizing the TBM excavated material in the asphalt mix designs or as a subbase for road construction requires some processing. He also mentioned that the TBM excavated material should be treated on site using mobile crusher equipment in order to be used as a non-structure concrete. However, using the TBM excavated material for backfilling was not demonstrated due to its high fine content and low hydraulic conductivity. Finally, the potential use of such material as a landfill liner was proposed with the requirement of additional sample crushing and the addition of some fine material such as bentonite. Voit and Kuschel [11], as well as and Voit et al. [12] examined the quality of the rock excavated from the longest underground railway line in the world, namely the Brenner Base Tunnel, and the treatment processes required to accomplish a high-quality level of recycled aggregates from such excavated material. After extensive research on the rock quality and concrete mix design, as well as on the appropriate implementation of the processing techniques and concrete mixing plants, they concluded that calcareous schists rock could be successfully recycled and processed as aggregate for shotcrete, structural, and inner-lining concrete, and as filter gravel. However, the quartz phyllite rock was found unsuitable as a concrete aggregate substitute due to intense foliation and mica-rich mineral composition.

### 1.2. Doha Metro Project: Background and Previous Studies

The Doha Metro project may represent the latest achievement in the field of TBM tunneling. It was entered into the Guinness Book of Records [13] due to its rapid construction, utmost safety considerations, and excellent quality standards. The Qatar Railway company constructed, using TBMs, 111 km of metro tunnels in less than two years. The Doha Metro tunnels included the construction, in several phases, of four main lines, namely Gold, Red, Green, and Blue. In the first phase, twenty-one TBMs were designed and their operations started in 2020. The second phase, which includes the excavation and construction of the Blue Line, is estimated to be completed in 2026. The excavated TBM materials of the first phase tunnels, which were about five million tons, were stored in four logistical areas in Doha. These areas were designated as MLPA South, MLPA West, MLPA North, and MLPA Al-Rifaa to differentiate between the characteristics of the TBM materials extracted from each area. Abu-Taqa et al. [14] studied the potential use of the TBM muck generated from Doha’s Metro Gold Line that was stockpiled at MLPA AL-Riffa in various construction applications. Laboratory tests were conducted on raw TBM material and the results obtained were compared with the specification limits established in the Qatari Construction Standards (QCS 2014) [15]. The raw TBM muck gradation complied with the QCS 2014 requirements for a fill material under buildings or road subgrades. However, the liquid limit and plasticity index of TBM muck were higher than the permissible limits. Conversely, TBM muck was not suitable as a concrete coarse aggregate substitute because its gradation did not meet the QCS 2014 requirements. Hence, physical processing is needed for TBM muck before using it as a concrete coarse aggregate substitute. The results obtained also showed that the raw TBM complies with the requirements of coarse aggregates, except for the water absorption and acid-soluble sulphate. Moreover, the results showed the presence of silica (Si) as a predominant component in TBM muck, which may cause higher Coefficient of Thermal Expansion (CTE) values. Consequently, this leads to a reduction in the compressive strength of the concrete mixes prepared using muck.

### 1.3. Research Objectives and Significance

As the properties of TBM muck are highly dependent on the geology of the tunneled area and the chemicals used in the TBMs facilitate the excavation, this study aims specifically at investigating the potential use of the Doha Metro Gold line TBM muck stockpiled at MLPA Al- Riffa as a partial or full replacement for concrete aggregates in Portland cement concrete (PCC). The properties of such TBM material have been investigated and reported by the authors in a previous work [14]. The results of this study may be applied for concrete mixtures prepared with TBM material that are extracted from different areas but have similar properties of Doha Metro’s TBM. It is worth noting that the raw TBM muck was subjected to physical modification (sieving) only to obtain sizes between 10 mm and 20 mm, and between 4 mm and 10 mm as per the QCS 2014 coarse aggregate requirements. The gabbro coarse aggregates used in the control mix were replaced with the raw TBM muck at percentages of 0%, 25%, 50%, and 75%. The fresh and hardened properties of the mixes were tested and compared to those of the control mix. It is worth noting that the hardened properties investigated in this study were limited to the compressive and flexure strengths. The elasticity and ductility should be also investigated by defining the stress-strain plots and future work should be directed towards this end. The results were used to decide the optimum TBM muck replacement percentage that may be utilized without a significant loss of the mechanical properties. The demonstration of the possibility of utilizing the TBM muck from the Doha Metro project excavations as well as using it as an alternative local aggregate, even partially, will lead to significant cost savings involved in importing aggregates from other countries, alongside environmental benefits related to the disposal of the stockpiled muck.

## 2. Experimental Program

Four mixes were prepared by replacing 0%, 25%, 50%, and 75% of gabbro coarse aggregates with TBM muck. The workability was determined for all mixes using the slump test. The mechanical properties were determined for all the mixes using flexural and compressive strength tests. The test results for TBM muck mixes were compared to those of the control mix (without TBM muck). Moreover, the fractured surfaces of the strongest samples of each mix were examined using SEM and EDX to understand their microstructures. Finally, the mechanical properties results were statically analyzed using *t*-tests.

### 2.1. Materials

Generally, Portland cement (CEM I) Class 42·5 R complying with EN 197-1 [16], natural-washed sand conforming to BS EN 12620 [17] (0–4 mm diameter), and tap water were used to prepare all concrete mixes. Gabbro aggregates and TBM muck of 4–10 mm and 10–20 mm diameters were used as coarse aggregates. The TBM muck, which was used in this study, was extracted from the Gold Line of the Doha Metro project. It was thoroughly investigated by the authors in an earlier study [14]. The TBM muck gradation was found to comply with the QCS 2014 [15] requirements for concrete coarse aggregates. Hence, the muck was classified into the acceptable sizes for coarse aggregates in concrete mixtures (10–20 mm and 4–10 mm). Table 1 summarizes the physical properties of TBM muck as concrete coarse aggregates [14].

Table 1 shows that many physical properties of TBM muck did not meet aggregate acceptable limits. Nevertheless, TBM muck was used herein to partially replace gabbro coarse aggregates to investigate the effect of such replacement on concrete’s strength. Its high-water absorption value may be the limiting factor for its use in concrete mixes. The high-water absorption and loss of magnesium sulphate soundness values of TBM muck may affect the frost and chemical resistance of the concrete mixture. However, this study was limited to the investigation of the strength of the TBM concrete mixture and future studies should be directed towards investigating other mixture properties, such as the freeze-thaw behavior, etc. To avoid concrete mix segregation and other associated concerns, a high-range water-reducer (HRWR), specifically CHRYSO Delta CQ 25, which conforms to ASTM C 494 Type-A, F and G [28], and BS EN 934-2 [29] standards, was incorporated into the mixes in various dosages based on the TBM muck replacement ratio. It is worth noting that cost of the HRWR used in this study is responsible compared to other similar products. As per the supplier, one liter of CHRYSO Delta CQ 25 costs USD 0.75, which is competitive among others. As per the manufacturer’s recommendations, the HRWR’s dosage shall not exceed the permissible one stated in the data sheet to avoid its adverse effect on fresh or hardened concrete properties. Table 2 summarizes the super-plasticizer’s properties and its recommended dosage.

### 2.2. Mixture Composition, Mixing, and Sample Preparation Procedures

Four different concrete mixes were designed to achieve a 28-day target cubic compressive strength of 40 MPa (approximately 32 MPa for cylindrical strength) with 0%, 25%, 50%, and 75% replacement of gabbro aggregates with TBM muck. As per Al-Ansary and Iyengar [30], the relative density of the imported gabbro (which is normally used for concrete production in Qatar) is about 2.8–2.9 and as per Table 1, the relative density of the TBM muck is about 2.75–2.8. Hence, the TBM replacement percentages based on the aggregate weight rather than volume could be feasible. The control mix (without TBM muck) had a water/cement ratio of 0.43 and a HRWR’s content of 0.82 L/100 kg of cementitious material. As previously noted and due to its high-water absorption, the water demand of the TBM muck mixes was higher than the control ones. However, QCS 2014 [15] limited the water/cement ratio to 0.5 for such a concrete grade. Hence, this value was not exceeded in all TBM muck mixes and the HRWR’s dosage was selected accordingly. The slump flow was checked immediately after mixing according to BS EN 12350-8 to achieve a target value of 160 ± 40 mm [31]. The HRWR’s dosage was adjusted gradually for the mixes with a slump value lower than the target one and the mix was repeated until achieving the target’s slump value. Table 3 presents the mix design and slump value for each mix. It should be noted that the HRWR’s dosages in the mixes containing 50% and 75% TBM muck replacement exceeded the manufacturer’s recommended values (i.e., 0.5 to 2.5 L per 100 kg of total cementitious material) to achieve the workability of the mix. The effect of this over-dosage on the hardened concrete properties should be noted and investigated.

Immediately after mixing, the slump flow value was measured according to BS EN 12350-8 [31]. Then, the specimens needed for the strength testing were cast. For each mix, 10 standard cylinders, according to ASTM C470 [32] (i.e.,152 mm in diameter and 304 mm in length), and 10 plain concrete beams with dimensions of 100 mm × 100 mm × 500 mm were casted. The specimens were demolded after 1 day and kept in the curing tanks until the day of testing.

### 2.3. Compressive and Flexure Strength Testing

Half of the samples were tested after 28 days of moist curing and the other half were tested after 56 days. The compressive and flexural strength testing of the samples was carried out according to ASTM C39 [33] and ASTM C78 [34], respectively. Then, *t*-statistical tests were used to determine whether there were any statistical differences between the strengths of the control mix and those containing the TBM muck.

### 2.4. Microstructural Analysis

From each mix, the fracture surface samples were collected from the strongest tested beam specimens. Scanning electron microscopy (SEM) and energy dispersive X-ray analysis (EDX) were used to investigate the microstructure and element composition, respectively, which may infer the mineralogy of the samples. It is worth noting that the EDX analysis was used herein to investigate the minerals present in the sample, which helps in justifying the strength behavior and other mixture properties (for example, thermal expansion). The SEM imaging started with vacuum drying of the samples followed by gold palladium coating in order to enhance the image resolution. A low-energy secondary and back-scattered electron were used to capture the SEM images at an accelerating voltage of 20 kV. The SEM images and EDX results were compared between the mixes.

## 3. Results and Discussion

### 3.1. Compressive Strength

The cylindrical compressive strength results at 28 and 56 days are presented in Table 4.

Table 4 shows that the compressive strength of the control mix was slightly enhanced by incorporating 25% of TBM muck. Strength increases of 4.8% and 6.1% were determined at 28 and 56 days, respectively. However, the compressive strengths for the mixes containing 50% TBM muck decreased by 10.4% and 11.5% at 28 and 56 days, respectively. Moreover, the compressive strengths for the mixes containing 75% TBM muck decreased by 16.5% and 14.4% at 28 and 56 days, respectively.

Additionally, *t*-statistical tests were used to draw a more accurate conclusion about the measured compressive strength results. The test null hypothesis assumed that the average strength values of both mixes are equal (i.e., µ_control mix_ = µ_s (TBM mix)_) and a two-tailed significance level of 0.05 (α = 0.05) was considered. The null hypothesis should be rejected if the *t*-statistic is larger than or equal to the critical *t*-test value. Table 5 shows the *t*-statistical analysis and the critical *t*-test values for the compressive strengths of all mixes at 28 and 56 days. It is worth noting that in the *t*-statistical test, the degree of freedom is dependent on the standard deviation and hence different degrees of freedom could be observed for the same sample size depending on the standard deviation of the results.

Table 5 shows that the enhancement of the compressive strength presented in Table 4 for Mix B (containing 25% TBM muck replacement) is statically insignificant. It shows that the reduction of the compressive strength in Mix C containing 50% TBM muck is also not significant. However, the reduction of the compressive strength due to the replacement of 75% of the gabbro aggregates with TBM muck is statistically significant.

The failure modes of the strongest specimen from each mix are presented in Figure 1a–d. As reported by Xu and Cai [35]**,** the crack pattern of the specimen under uniaxial compression depends on its microstructure and stress state. The fracture patterns shown in Figure 1a–d are similar to those specified in ASTM C39 [33] for the cylinders under uniaxial compressive loads. The figures show that the samples from Mix D (containing 75% TBM muck replacement) split vertically after failure, which indicates a brittle failure under uniaxial compression. However, the cylinders of the control Mix A, Mix B, and Mix C (which contained 0%, 25%, and 50% TBM muck replacement, respectively) failed along inclined shearing planes, which is considered more of a ductile shear failure rather than a brittle splitting one. It could be also concluded that as the TBM muck content increased above 50% of the weight of the coarse aggregates in the mix, it tended to have a brittle failure pattern under uniaxial compression loads and hence no apparent deformation may be witnessed before fracture. This may indicate a degradation in the mixture’s ductility.

### 3.2. Flexural Strength

Table 6 presents the flexural strength results for the samples at 28 and 56 days.

The results showed that the flexural strength decreased due to the partial replacement of coarse aggregates with TBM muck, even for small replacement percentages. This finding does not correspond with the increased compressive strength results for Mix B containing 25% TBM muck as a coarse aggregate replacement. Table 6 shows that the flexural strength decreased by 9% and 21.2% at 28 days and by 15% and 24.2% at 56 days for Mixes B and C, respectively. However, the flexural strength decreased by only 3.4% at 28 days and 17.3% at 56 days for Mix D containing 75% TBM muck replacement. The muck material’s heterogeneity may be one reason for these inconsistent results. However, the analysis of the fractured samples’ microstructures will be analyzed in the following section to draw a better explanation.

A *t*-statistical analysis was also carried out to confirm the extent of the significance of the flexural strength reduction. Table 7 summarizes the *t*-statistical analysis for all mixes at 28 and 56 days.

Table 7 shows that the reduction in the 28th day of the flexural strength may be considered significant for Mix C containing 50% TBM muck. However, the reduction in the flexural strength becomes significant for all mixes at 56 days. Thus, replacing concrete coarse aggregates with untreated muck, even with small percentages, may adversely affect the flexural strength, especially in the long-term.

### 3.3. Microstructural Analysis

The EDX analysis of the fractured surfaces for the strongest beam samples at 28 days defines their element composition, which may infer the mineralogy as shown in Figure 2, Figure 3, Figure 4 and Figure 5. Moreover, the SEM images were taken from the fractured surfaces (scale of 2 µm) to investigate their microstructures (Figure 6a–d).

The EDX graphs show that Mixes B, C, and D contained sodium ions (Na^+^) in contrast with the control Mix A. Moreover, the calcium (Ca^++^) ions’ content in Mixes B, C, and D were less than the Ca^++^ content in Mix A. The muck Na^+^ ions’ source may be the rock, which is being excavated, or the chemical additives used to facilitate the excavation process. The presence of sodium (Na^+^) ions in the mix may justify the strength’s loss as witnessed in the samples containing TBM muck, especially in the long-term. Wang et al. [36] studied the effects of sodium bicarbonate and sodium carbonate as concrete accelerators on the hydration process and on the Portland cement paste hardened properties. They reported that both accelerators, with an optimum concentration of 1% by weight of the Portland cement, may shorten the paste’s setting time and increase the early strength without having a detrimental effect on its long-term strength. However, an increase of both accelerators above 1% by weight could decrease the paste’s long-term strength. The strength reduction may be attributed to the fact that Na^+^ ions may replace the Ca^++^ in the C-S-H gel at a later age, causing discontinuities of the gel and hence reducing the strength. This conclusion may be also applicable for the mixes containing TBM muck due to the presence of the Na^+^ in the muck material and justifies the significant strength reduction at 56 days.

The SEM results for all mixes at 28 days are shown in Figure 6a–d. The images show that the needle-shaped ettringite and carbon hydroxide (C-H) contents in the TBM muck mixes were higher than those in the control mix. It is worth mentioning that the C-S-H gel in the TBM muck mixes was less than that in the control mix. While the early age strength is mainly influenced by both ettringite and the C-S-H gel, the later age strength is mainly influenced by the C-S-H gel’s microstructure. These results agree with the flexural strength results at a later age, as presented in Table 5.

To reduce their adverse effects on the C-S-H microstructure and mixture strength, the Na^+^ content must be reduced in TBM muck when used in concrete. The use of pozzolanic materials such as fly ash, silica fume, etc., may be useful. Further research work will be needed using a stabilization technique.

## 4. Conclusions

This study investigated the potential use of TBM muck obtained from the Doha Metro project’s Gold Line as a partial replacement for gabbro aggregates. In addition to the control mix (without TBM muck), TBM muck mixes with 25%, 50%, and 75% replacement of coarse aggregates were prepared. The slump test was performed on all mixes to ensure good workability. The compressive and flexural strengths were determined for all mixes at 28 and 56 days, and were compared to those of the control mix. Moreover, EDX analysis and SEM images were taken in order to study the microstructure and element composition, which may infer the mineralogy of the mixes and analyze the results. The following conclusions can be drawn:Due to TBM muck’s high-water absorption, the amount of water needed to achieve a sufficient mixture’s workability was increased. Hence, a high-range water-reducer (HRWR) was added with different dosages to achieve standard slump values. Increasing the HRWR dosages above the recommended limits could affect the strain gain.The compressive strength was slightly better for the 25% TBM muck mix in comparison with the control at 28 and 56 days. However, the *t*-statistical analysis showed that this improvement was statically insignificant.Increasing the TBM muck’s content beyond 25% reduced the compressive strength of the mixes at 28 and 56 days. This reduction was significant for the 75% TBM muck mix.The failure modes under uniaxial loads showed that by increasing TBM muck content above 50%, the samples tend to exhibit columnar brittle failure and hence no apparent deformation may be witnessed before fracture. Accordingly, the mixture’s ductility may be decreased by increasing the TBM content above 50% of coarse aggregates.The use of TBM muck in concrete mixtures had an adverse effect on the flexural strength, especially in later ages. The *t*-statistical analysis showed that the flexural strength reductions in TBM muck concrete mixes were statically significant and should be taken into consideration.The EDX analysis showed that the mixes prepared with TBM muck contained Na^+^ ions, which may be the reason for the strength decay observed, especially at later ages. Thus, it is recommended to investigate the use of fly ash, silica fume, etc., in TBM muck–concrete mixes.The SEM images showed that the ettringite and carbon hydroxide (C-H) contents in TBM muck mixes were higher than those in the control mix, while the C-S-H gel in the TBM muck mixes was less. This may justify the reduction of the flexural strength in TBM muck–concrete mixes.

According to the aforementioned observations, it can be concluded that the TBM muck could be utilized into concrete mixtures without a significant loss of strength; however, physical processing and chemical stabilization may be needed to enhance the properties of the TBM muck and to avoid the adverse effect of its composition on the mechanical properties of concrete mixtures. This should be the subject for future study to demonstrate the potential utilization of TBM muck for construction applications.

## Figures and Tables

**Figure 1 materials-14-06263-f001:**
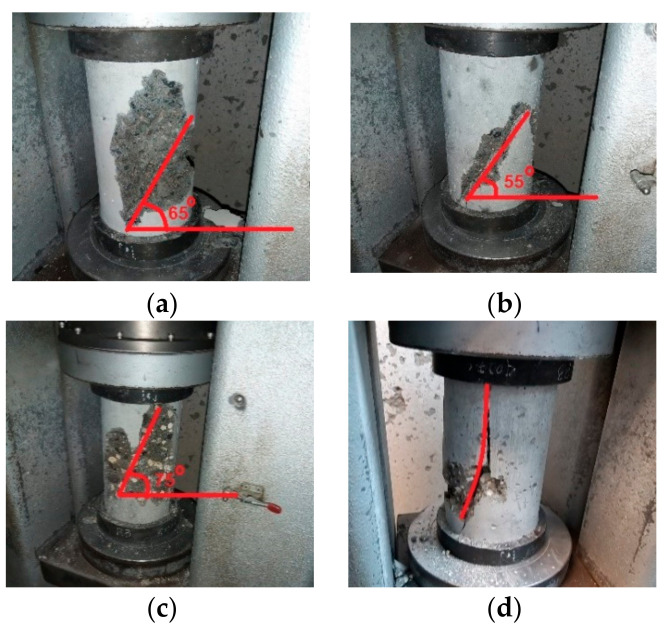
Failure modes of the strongest cylinder specimens under compression at 28 days: (**a**) Mix A; (**b**) Mix B; (**c**) Mix C; and (**d**) Mix D.

**Figure 2 materials-14-06263-f002:**
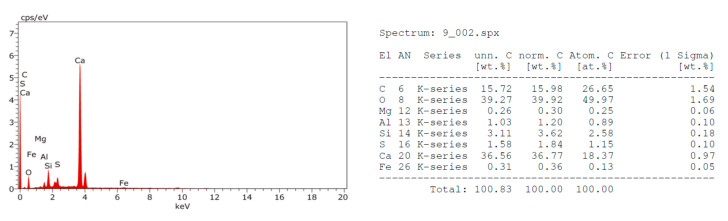
EDX analysis of Mix A fractured surface.

**Figure 3 materials-14-06263-f003:**
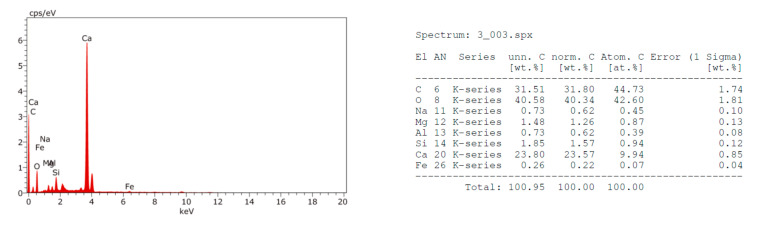
EDX analysis of Mix B fractured surface.

**Figure 4 materials-14-06263-f004:**
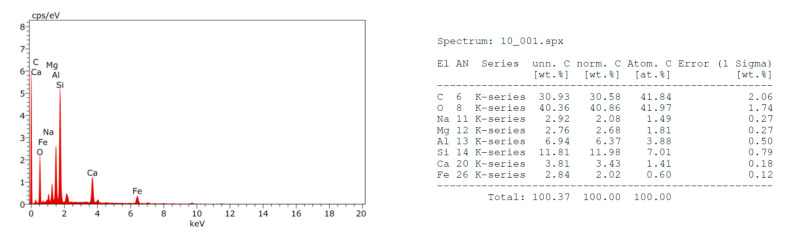
EDX analysis of Mix C fractured surface.

**Figure 5 materials-14-06263-f005:**
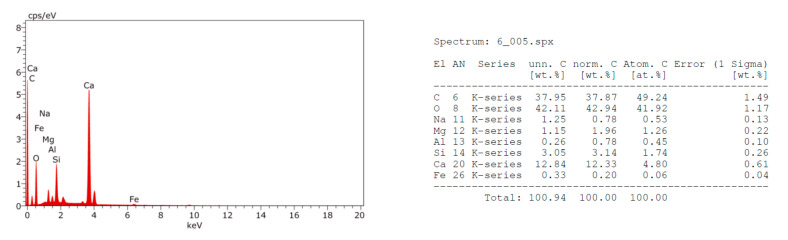
EDX analysis of Mix D fractured surface.

**Figure 6 materials-14-06263-f006:**
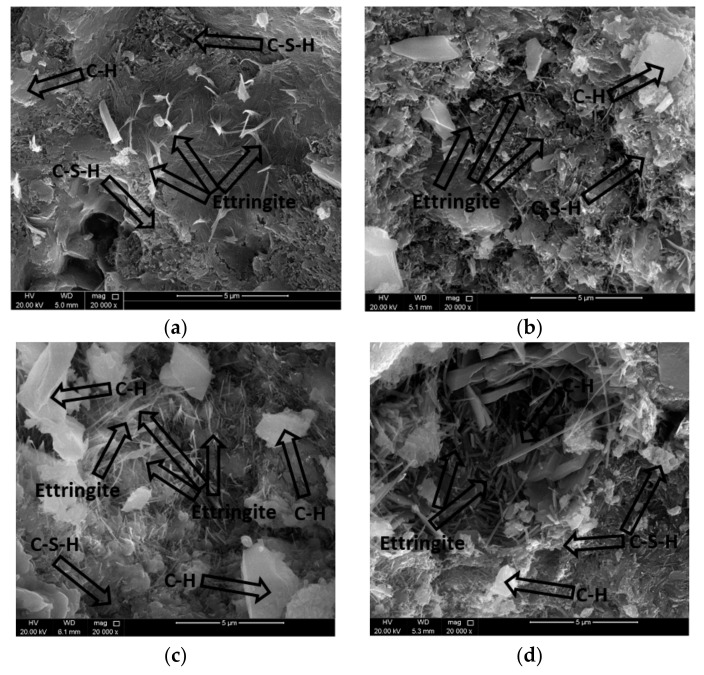
SEM images of the fractured surface of the strongest beam samples at 28 days: (**a**) Mix A; (**b**) Mix B; (**c**) Mix C; and (**d**) Mix D.

**Table 1 materials-14-06263-t001:** Properties of TBM muck as concrete coarse aggregates.

Test	Test Standard	Average Test Result		QCS 2014 Limits
		4–10 mm	10–20 mm	
Particle relative density	BS/EN 1097-6 [18]	2.80	2.75	Min. 2.0
Water absorption (%)	BS/EN 1097-6 [18]	7.73	7.07	Max. 2.0
Clay lumps and friable particles (%)	ASTM C 142 [19]	2.80	2.05	Max. 2.0
Shell content (%)	BS/EN 933-7 [20]	Nil	Nil	Max. 3.0
Flakiness index (%)	BS/EN 933-3 [21]	4.33	8.33	Max. 35
Aggregate drying shrinkage (%)	BS/EN 1367-4 [22]	0.03	0.03	Max. 0.075
Loss by magnesium sulphate soundness (%)	BS/EN 1367-2 [23]	53.67	50.67	Max. 15
Loss by Los Angeles abrasion (%)	BS/EN 1097-2 [24]	40.00	41.00	Max. 30
Acid-soluble chloride (%)	BS/EN 1744-5 [25]	0.020	0.020	Max. 0.03
Acid-soluble sulphate (%)	BS/EN 1744-1 [26]	1.30	1.20	Max. 0.3
Light-weight particles (%)	ASTM C123 [27]	<0.1	<0.1	Max. 0.5

**Table 2 materials-14-06263-t002:** Properties of the high-range water-reducer CHRYSO Delta CQ 25.

Appearance	Pale yellow liquid
Density	1.055 ± 0.020
pH value	4.50 ± 2.00
Chloride ion content	Nil
Recommended dosage	0.5 to 2.5 L per 100 kg of total cementitious material.

**Table 3 materials-14-06263-t003:** Mix designs and the slump flow values of all mixes.

Constituent Material	Mix A (0% TBM)	Mix B (25% TBM)	Mix C (50% TBM)	Mix D (75% TBM)
Cement, (kg/m^3^)	390.00	390.00	390.00	390.00
Gabbro coarse aggregate, 10–20 mm (kg/m^3^)	674.00	505.50	337.00	168.50
TBM coarse aggregate, 10–20 mm (kg/m^3^)	_____	168.50	337.00	505.5
Gabbro coarse aggregate, 4–10 mm (kg/m^3^)	317.00	237.75	158.50	79.25
TBM coarse aggregate, 4–10 mm (kg/m^3^)	_____	79.25	158.50	237.75
Fine aggregate (natural-washed sand) (kg/m^3^)	889.00	889.00	889.00	889.00
Water (kg/m^3^; W/C)	167.70 (0.43)	180.00 (0.46)	188.00 (0.48)	192.00 (0.49)
HRWR, CHRYSO Delta CQ 25 (liters/100 kg)	0.80	2.5	4.73	5.95
Slump flow (mm)	160	161	170	172

**Table 4 materials-14-06263-t004:** Compressive strength values of all mixes at 28 and 56 days.

Mix	Sample No.	28 Days		56 Days	
		Compressive Strength (MPa)	Average Value (MPa)	Compressive Strength (MPa)	Average Value (MPa)
Mix A (control)	1	38.7	38.0	37.1	38.6
	2	36.7		35.4	
	3	40.8		41.2	
	4	35.2		42.1	
	5	38.4		37.4	
Mix B (25%TBM)	1	39.3	39.8	39.5	41.0
	2	40.3		41.5	
	3	40.4		41.1	
	4	40.0		42.6	
	5	39.0		40.3	
Mix C (50%TBM)	1	32.9	34.0	28.9	34.2
	2	36.7		33.1	
	3	37.0		37.5	
	4	37.3		36.9	
	5	26.3		34.6	
Mix D (75%TBM)	1	33.1	31.7	29.4	33.1
	2	31.7		34.4	
	3	32.8		30.9	
	4	28.3		32.9	
	5	32.7		37.7	

**Table 5 materials-14-06263-t005:** Compressive strength *t*-statistical analysis.

	Mix	Degree of Freedom (DF)	Standard Error (SD)	*t*-Statistic Value	Critical *t*-Test Value	Remarks
28 days	Mix B (25% TBM)	5	0.99	1.86	2.571	Do not reject null hypothesis (insignificant enhancement)
	Mix C (50% TBM)	6	2.30	1.71	2.447	Do not reject null hypothesis (insignificant reduction)
	Mix D (75% TBM)	8	1.30	4.80	2.306	Reject null hypothesis (significant reduction)
56 days	Mix B (25% TBM)	5	1.39	1.70	2.571	Do not reject null hypothesis (insignificant enhancement)
	Mix C (50% TBM)	8	2.01	2.21	2.306	Do not reject null hypothesis (insignificant reduction)
	Mix D (75% TBM)	8	1.93	2.89	2.306	Reject null hypothesis (significant reduction)

**Table 6 materials-14-06263-t006:** Flexural strength results for all mixes at 28 and 56 days.

Mix	Sample No.	28 Days		56 Days	
		Flexural Strength (MPa)	Average Value (MPa)	Flexural Strength (MPa)	Average Value (MPa)
Mix A (control)	1	6.1	5.4	5.6	6.1
	2	5.3		6.5	
	3	5.2		6.3	
	4	4.9		5.6	
	5	5.3		6.6	
Mix B (25%TBM)	1	4.2	4.9	4.9	5.2
	2	5.4		5.9	
	3	5.1		5.4	
	4	4.9		4.5	
	5	4.8		5.3	
Mix C (50%TBM)	1	3.0	4.2	5.0	4.6
	2	3.9		5.0	
	3	4.0		4.1	
	4	5.1		4.6	
	5	5.1		4.5	
Mix D (75%TBM)	1	5.3	5.2	4.9	5.1
	2	5.4		4.9	
	3	4.5		6.1	
	4	5.1		4.1	
	5	5.6		5.3	

**Table 7 materials-14-06263-t007:** Flexural strength *t*-statistical analysis.

	Mix	Degree of Freedom (DF)	Standard Error (SD)	*t-*Statistic Value	Critical *t*-Test Value	Remarks
28 days	Mix B (25% TBM)	8	0.28	1.71	2.306	Do not reject null hypothesis (insignificant reduction)
	Mix C (50% TBM)	6	0.45	2.56	2.447	Reject null hypothesis (significant reduction)
	Mix D (75% TBM)	8	0.27	0.66	2.306	Do not reject null hypothesis (insignificant reduction)
56 days	Mix B (25% TBM)	8	0.32	2.86	2.306	Reject null hypothesis (significant reduction)
	Mix C (50% TBM)	8	0.28	5.37	2.306	Reject null hypothesis (significant reduction)
	Mix D (75% TBM)	7	0.39	2.71	2.365	Reject null hypothesis (significant reduction)

## Data Availability

The data used to support the findings of this study are available upon request from the corresponding author.

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
