# Peer review of "Performance of Concrete Mixes Containing TBM Muck as Partial Coarse Aggregate Replacements"

_materials, 2021, doi:10.3390/ma14216263_

Round 1

Reviewer 1 Report

General comment:

This paper investigated the performance of concrete with the coarse aggregate replacement by TBM muck. The mechanical properties and microstructures have been studied in this paper. However, there are some suggestions that the authors may consider improving the quality of this manuscript.

Technique comments:

  1. From line 154-161, It can be noticed that the use of HRWR has been significantly increased when the TBM was involved in the mixtures. However, the cost of HRWR is much higher than the cost of coarse aggregate. What do you think is the cost-performance of the TBM muck concrete if you need to increase the dosage of HRWR?

  1. Line 150, please check the line number format, which is not correct as shown in the paper.

  1. Table 1, the absorption of water of TBM muck is significantly higher than the conventional aggregate materials as required by the standard that you listed. The increased absorption is related to the high air voids content in the aggregate. Thus, the increased air voids content may also affect the freeze-thaw durability of the concrete materials. At the same time, the loss by Magnesium Sulphate soundness is also higher than that of coarse aggregate.

How do you think about the frost resistance and chemical resistance of the TBM muck concrete material that you proposed in this paper? Those properties are critical for concrete materials since durability is also a very important factor when you innovate a new concrete material.

  1. For the concrete mixture as shown in Table 3, the authors replaced the coarse aggregate based on the weight of the aggregate. However, the density is different by comparing with the normal coarse aggregate. Thus, the mixtures that you prepared resulted in different volumes, but generally, people design the mixture based on a constant volume (per m3). Why did you replace the aggregate in volume content to make sure the volume be consistent?

  1. Also, please make the unit of mixture design in Table 3 as kg/m3.

  1. Table 4 and Figure 2 are repeated, as they represented the same results.
  2. Line 244-248, a more brittle failure relates to low compressive strength? Generally, when compressive strength increases, the brittleness would also increase.

  1. Line 285, check the format.

  1. The conclusion is too long. Some of them can be combined as one item, not that many conclusions can be obtained from this study.

Reviewer 2 Report

The article is about the performance of concrete mixes containing TBM muck as partial coarse aggregate replacement. However, some issues must to be addressed:

  1. Introduction section use references very old. Please rewrite introduction taking into consideration scientific literature from last 3 years.
  2. Figure 1 can be removed: has no scientific meaning.
  3. Discussion section doesn’t take into consideration latest references.
  4. EDX analysis offer only qualitative information abut composition: is better to change analysis with one which offer the chemical composition of the mixes, because nobody understand what kind of raw materials were used (their properties) and what kind of materials were obtained (their properties, including chemical composition!).
  5. Line 326: EDX analysis doesn’t offer information about microstructure.
  6. Conclusion section must to be synthesized.

Reviewer 3 Report

The presented manuscript is devoted to the potential utilization of the tunnel boring machines muck as a partial or full replacement for concrete aggregates in portland cement concrete. 

In the work the authors prepared mixes with different partial muck replacement of coarse aggregates and tested their properties. In my opinion the described results are interesting and valuable and they have application value. Therefore I believe the paper is worth publishing. 

I believe the resubmitted manuscript has been sufficiently improved. In my opinion it is suitable for publication in the present form.

Author Response

The authors would like to thank the  reviewer for his comments and suggestions, which definitely improved the quality of our manuscript. The reviewer has accepted the manuscript in the current format. 

Reviewer 4 Report

This is a very interesting paper, which also deals with the highly topical issue of sustainable development. The reviewer would like to suggest only two minor corrections.

The first suggestion concerns the Introduction section, as it is very long compared to the length of the other sections. The authors could shorten this section by moving information on the Doha Metro project (starting from line 107) in a new section or sub-section.

The second observation is more important as regards the contents. In the fourth item of the bulleted list, the Authors state: “increasing TBM muck content 356 increased the failure plane angle until it reached approximately 90° for the 75% TBM muck 357 mix”. The Authors should describe this behavior in more detail, since, according to Figure 1, the failure plane angle of Mix B is less than that of Mix A and only begins to increase for a higher content of TBM muck. Therefore, the failure plane angle does not increase monotonically with the content of TBM muck, as it might seem when reading this sentence.

Reviewer 5 Report

Comments

This paper studied the performance of concrete mixes containing TBM muck as partial coarse aggregate replacement. The outcome of the paper is interesting however, there are several aspects that need to be improved. The reviewer can only recommend for publication if the author satisfactorily address the following major comments in the revised version.

  1. The author need to plot the stress strain curve for the concrete tests.
  2. The research questions and justification of selected parameters should be highlighted.
  3. Which test standards was considered in this study? 
  4. The deterioration mechanism of the specimen should be discussed more clearly.
  5. The novelty of the study should be highlighted more clearly at the end of introduction section. How this study is different from the published study in literature?
  6. How the outcome of this study will benefit researchers and end users? This need to be highlighted in introduction or end of conclusion.
  7. The importance of coarse aggregate replacement and the recent investigation in this area should be discussed in introduction section to improve the background study. Recently, several alternative materials used to replace coarse aggregates [Ref: Recycling of landfill wastes (tyres, plastics and glass) in construction–A review on global waste generation, performance, application and future opportunities] and [Ref: Investigation on the physical, mechanical and microstructural properties of epoxy polymer matrix with crumb rubber and short fibres for composite railway sleepers]. Suggest to include them in introduction section with proper citations to improve the background study.

I would be happy to see the revised version to understand how these comments are being addressed.

Round 2

Reviewer 1 Report

The comments have been clearly addressed, it could be accepted in its current format.

Author Response

The authors would like to thank the  reviewer for his comments and suggestions, which definitely improved the quality of our manuscript.

Reviewer 2 Report

The EDX analysis is used to determine "the presence 23 of Na+ ions", which is not possible. On the entire article the authors use the EDX analysis to sustain different affirmations without to be justified. This kind of analysis cannot be a support for ... line 345 "minerology". If the authors insist to use this analysis in this way, inside the paper, then article must to be rejected.

Reviewer 5 Report

Comment 1 and comment 7 were not addressed properly. 

Round 3

Reviewer 2 Report

I am very strong in my position about "element composition". May be is about "elemental composition", but EDX analysis is only a qualitative analysis and cannot put in evident a chemical element in orider to speculate furthoer more that will be later replaced ...

The analysis must to be replaced with corresponding one in article.